# RamanSPy: Augmenting Raman Spectroscopy Data Analysis with AI

**Dimitar Georgiev** [1] [2]  **Simon Vilms Pedersen** [2]  **Ruoxiao Xie** [2]  **Álvaro Fernández-Galiana** [2]  **Molly M. Stevens** [2]
**Mauricio Barahona** [3]

## Abstract

Raman spectroscopy is a non-destructive and label-free chemical analysis technique, which plays a key role in the analysis and discovery cycle of various branches of life and material sciences. Recently, there has been a marked increase in the adoption of machine learning techniques in Raman spectroscopic analysis. Nonetheless, progress in the area is still impeded by the lack of software, methodological and data standardisation, and the ensuing fragmentation and lack of reproducibility of analysis workflows thereof. To address these issues, we introduce *RamanSPy*, an open-source Python package for Raman spectroscopic data analysis, which supports day-to-day tasks, integrative analyses, the development of methods and protocols, and the integration of advanced data analytics. *RamanSPy* is highly modular, not tied to a particular technology or data format, and can be readily interfaced with the burgeoning ecosystem for data science, statistical analysis and machine learning in Python. *RamanSPy* is hosted at https://github.com/barahona-research-group/RamanSPy, supplemented with extended online documentation, available at https://ramanspy.readthedocs.io, that includes tutorials, example applications, and details about the real-world research applications presented in this paper.

---

[1]Department of Computing & UKRI Centre for Doctoral Training in AI for Healthcare, Imperial College London, London, United Kingdom, SW7 2AZ [2]Department of Materials, Department of Bioengineering & Institute of Biomedical Engineering, Imperial College London, London, United Kingdom, SW7 2AZ [3]Department of Mathematics, Imperial College London, London, United Kingdom, SW7 2AZ. Correspondence to: Molly M. Stevens <m.stevens@imperial.ac.uk>, Mauricio Barahona <m.barahona@imperial.ac.uk>.

*Accepted at the 1st Machine Learning for Life and Material Sciences Workshop at ICML 2024.* Copyright 2024 by the author(s).

## 1. Introduction

Raman spectroscopy (RS) is a powerful sensing modality based on inelastic light scattering, which provides qualitative and quantitative chemical analysis with high sensitivity and specificity (Colthup, 2012). RS yields a characterisation of the vibrational profile of molecules, which can help elucidate the composition of chemical compounds, biological specimens and materials (McCreery, 2005; Shipp et al., 2017; Fernandez-Galiana et al., 2023). In contrast to most conventional technologies for (bio)chemical characterisation (e.g., staining, different omics, fluorescence microscopy and mass spectrometry), RS is both label-free and non-destructive, thereby allowing the acquisition of rich biological and chemical information without compromising the structural and functional integrity of probed samples. This advantage has enabled a broad range of applications of RS across biomedical and pharmaceutical research (Smith et al., 2016; Shipp et al., 2017; Cialla-May et al., 2017; Wang et al., 2018; Vankeirsbilck et al., 2002; Auner et al., 2018; Kong et al., 2015), materials science (Weber & Merlin, 2013; Kumar, 2012), environmental science (Halvorson & Vikesland, 2010; Ong et al., 2020), and others (Pang et al., 2016; Chalmers et al., 2012; Terry et al., 2022).

An area of topical interest is the frontier of Raman spectroscopy, chemometrics and artificial intelligence (AI), with its promise of more autonomous, flexible and data-driven RS analytics (Pan et al., 2022; Luo et al., 2022; Lussier et al., 2020). There has been a recent surge in the adoption of AI methods in Raman-based research (Fernandez-Galiana et al., 2023), with applications to RS now spanning domains as broad as the identification of pathogens and other microbes (Ho et al., 2019; Lu et al., 2020; Yan et al., 2021; Wang et al., 2021); the characterisation of chemicals, including minerals (Carey et al., 2015), pesticides (Zhu et al., 2021) and other analytes (Han & Ram, 2020; Akpolat et al., 2020); the development of novel diagnostic platforms (Ralbovsky & Lednev, 2020; Talari et al., 2019; Heng et al., 2021; Zhang et al., 2022); as well as the application of techniques from computer vision for denoising and super-resolution in Raman imaging (Horgan et al., 2021).

Yet, progress in the area is still hindered by practical factors stemming from the restrictive, functionally disparate,

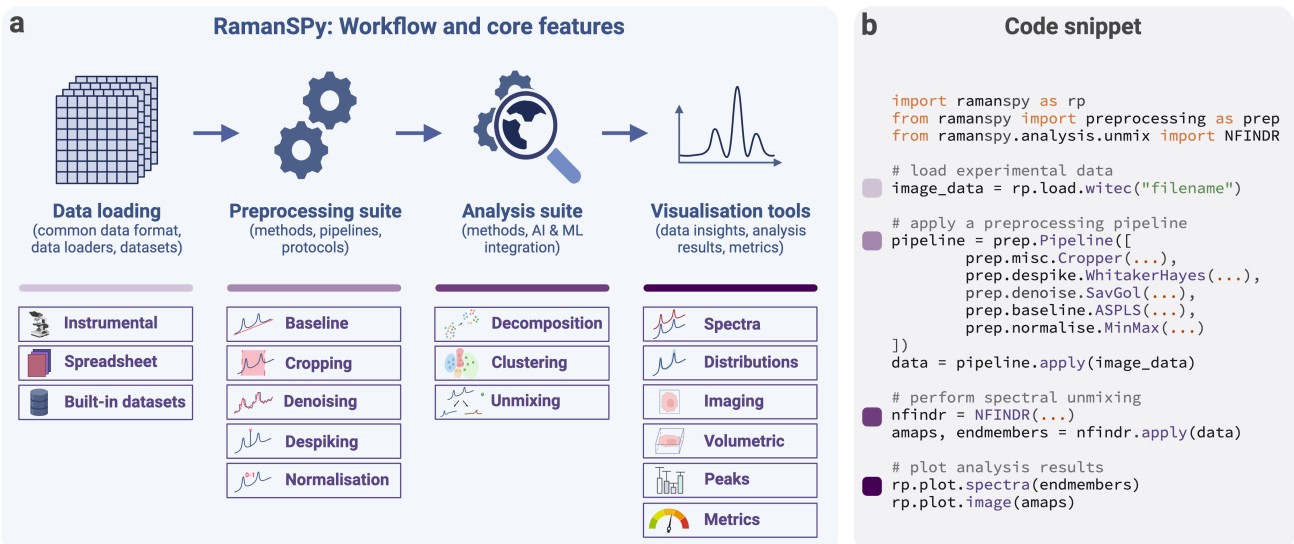

**Figure 1.** Core infrastructure of *RamanSPy*. a, *RamanSPy* provides a comprehensive library of standardised, simple-to-use procedures for data loading, preprocessing, analysis and visualisation organised within a modular and extensible architecture. b, An example workflow use case in *RamanSPy*: Raman data is loaded, preprocessed and analysed in a few lines of code.

and highly encapsulated nature of current commercial software for RS data analysis. RS data analysis often operates within proprietary software environments and data formats, which have induced methodological inconsistencies and reduced cross-platform and integrative efforts, with growing concerns around reproducibility (Byrne et al., 2016; Tanwar et al., 2021; Barton et al., 2022; Möller et al., 2017; Ntziouni et al., 2022). These restrictions have hampered the development, validation and deployment of emerging AI-based technologies for RS.

In response to these challenges, we have developed *RamanSPy* - a modular, open-source framework for Raman Spectroscopy analytics in Python. *RamanSPy* is designed to systematise day-to-day workflows, enhance algorithmic development, integration and reproducibility, and accelerate the adoption of AI technologies into the RS field.

## 2. Core infrastructure of RamanSPy

*RamanSPy* is based on a modular, object-oriented programming infrastructure comprising a comprehensive collection of pre-defined tools for RS data analysis, which streamlines the analysis life cycle and reduces computational barriers to RS analytics (Figure 1).

**Data loading and management.** The framework adopts a scalable array-based data representation based on *NumPy* (Harris et al., 2020), which accommodates different spectroscopic modalities, including single-point spectra, Raman imaging data, and volumetric scans. Experimental data can be loaded through custom loaders built into *RamanSPy*

or through standard tools available in Python. The data representation functions as a common data container that facilitates the integrative analysis of data across setups and vendors, independent of instrumental origin and acquisition modality, and defines the interface between RS data and analysis tools within and beyond *RamanSPy*.

**Preprocessing, analysis and visualisation.** On top of its data management infrastructure, *RamanSPy* provides an extensive toolbox for preprocessing, analysis and visualisation. The preprocessing suite includes techniques for denoising, baseline correction, cosmic spike removal, normalisation and background subtraction, among others. Likewise, the analysis toolbox includes modules for decomposition (useful for dimensionality reduction), clustering and spectral unmixing. *RamanSPy* also includes a set of data visualisation tools, intended to facilitate routine visualisation and exploratory analysis. All these modules are organised into a common class structure, which standardises their application across projects and datasets to facilitate transferable analysis workflows. Note that this suite is highly flexible and designed to cater to a wide range of requirements, applications and user profiles.

**Automated pipelining of spectral preprocessing protocols.** Due to a lack of standardisation and frameworks that streamline the preprocessing of RS data (Byrne et al., 2016), researchers tend to utilise variable preprocessing protocols, often dispersed across different software systems (Rozenstein et al., 2014; Alshdaifat et al., 2021). This significantly affects reproducibility and validation, especially in the con-

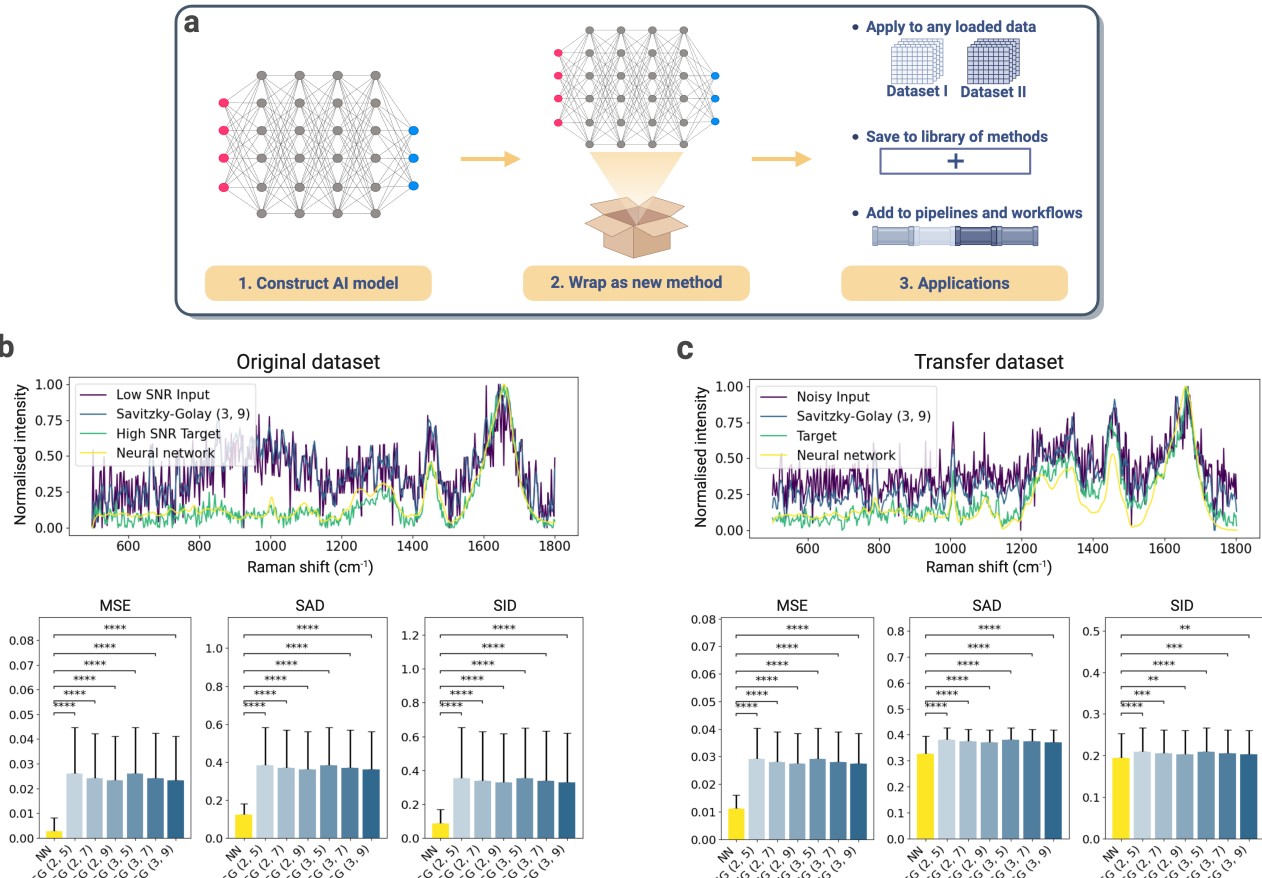

*Figure 2. RamanSPy* interfaces with AI/ML Python frameworks to create new methods for RS analysis. a, *RamanSPy* allows users to incorporate AI/ML models seamlessly into pipelines created within the platform. b, A pre-trained 1D ResUNet deep-learning denoiser (Horgan et al., 2021) is integrated as a preprocessing module within *RamanSPy* to investigate its performance against the Savitzky-Golay (SG) filter (Savitzky & Golay, 1964). top, Denoising of a spectrum from Horgan et al. (2021), where the low-SNR (purple) is the input and the high-SNR (green) is the target. The data is denoised with an SG filter of polynomial order 3 and kernel size 9, SG(3, 9) (blue), and with the implemented deep-learning denoiser (yellow). bottom, The results on the test set from Horgan et al. (2021) ($n = 12694$) show that the deep-learning denoiser outperforms six SG filters across three performance metrics (MSE, SAD, SID). Error bars represent one standard deviation around the sample mean. Statistical significance measured with a two-sided Wilcoxon signed-rank test with adjustment for multiple comparisons based on Benjamini-Hochberg correction (Benjamini & Hochberg, 1995) (* $P < 0.05$, ** $P < 0.01$, *** $P < 0.001$, **** $P < 0.0001$). c, Same analysis on unseen data from Kallepitis et al. (2017) ($n = 1600$). The input (purple) corresponds to data contaminated with added noise and the target (green) to the original data. In this case, the deep-learning denoiser only shows an improvement for MSE.

text of AI model development.

To facilitate the creation of reproducible protocols, *RamanSPy* incorporates a pipelining infrastructure, which systematises the process of creating, customising and executing preprocessing pipelines. Users can use a specialised class, which defines a generic, multi-layered preprocessing procedure, to assemble pipelines from selected built-in preprocessing modules or other in-house methods. To reduce overhead, constructed pipelines are designed to function exactly as any single method, i.e., they are fully compatible with the rest of the modules and data structures in the package. Furthermore, pipelines can be easily saved, reused and

shared (e.g., upon publication) to foster the development of a shared repository of preprocessing protocols. As a seed to this repository, *RamanSPy* provides a library of assembled preprocessing protocols (custom pre-defined, or adapted from the literature (Bergholt et al., 2016)), which users can access and exploit.

## 3. AI and Raman Spectroscopy: Bridging the gap with *RamanSPy*

To help accelerate the adoption of AI technologies for RS analysis, *RamanSPy* is endowed with a permeable archi-

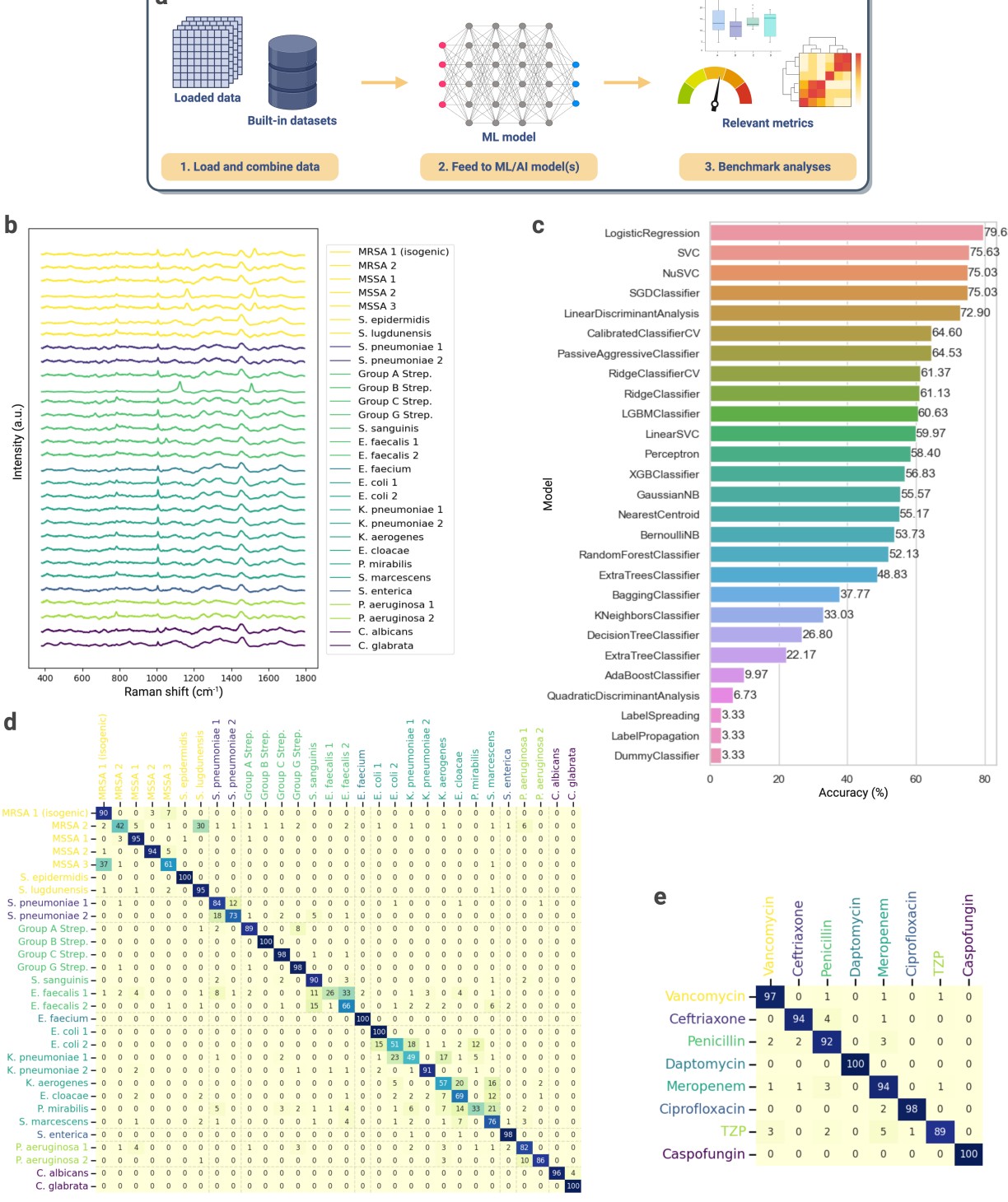

Figure 3. *RamanSPy* as a suite for algorithmic development. a, Data representations in *RamanSPy* are compatible with the Python AI/ML ecosystem, allowing data flow from *RamanSPy* to *scikit-learn* (Pedregosa et al., 2011), *PyTorch* (Paszke et al., 2019), *tensorflow* (Abadi et al., 2016), etc. *RamanSPy* is also equipped with standard datasets and relevant metrics to support model development and validation. b-e, Benchmarking ML classification models on the task of bacteria identification using Raman spectra from Ho et al. (2019). b, Mean Raman spectra of each bacterial species in the dataset used for training. Spectra are min-max normalised to the range 0–1 for visualisation purposes. c, Benchmarking results of 28 ML models. The best accuracy was achieved by the logistic regression classifier. d-e, Confusion matrices for the best species-level (d) and antibiotic-level (e) classifier with accuracies of 79.63% and 94.63%, respectively.

tecture that streamlines the interface between Raman spectroscopic data and the burgeoning machine learning (ML) ecosystem in Python. This is complemented by access to built-in datasets and performance metrics to further support the development and testing of new models and algorithms. We show below two examples of *RamanSPy*'s capabilities for ML integration and validation.

**AI integration.** First, *RamanSPy* allows the seamless integration of methods from standard AI/ML frameworks in Python (e.g., *scikit-learn* (Pedregosa et al., 2011), *PyTorch* (Paszke et al., 2019) and *tensorflow* (Abadi et al., 2016)) as tools for RS analysis (Figure 2a). As an illustration of how custom methods can be integrated into analysis pipelines within *RamanSPy*, we use our package to construct a deep learning denoiser based on the one-dimensional ResUNet model - a fully convolutional UNet neural network with residual connections, presented in Horgan et al. (2021). To do this, we simply wrap within *RamanSPy* the pre-trained neural network[1] as a custom denoising method. Once wrapped, the denoiser is automatically compatible with the rest of *RamanSPy* and can be readily employed for different applications. For instance, we replicated the results in Horgan et al. (2021), and show in Figure 2b that the application of this deep-learning denoiser to the low signal-to-noise ratio (SNR) test set from Horgan et al. (2021) consistently outperforms the commonly-used Savitzky-Golay filter (Savitzky & Golay, 1964). This is quantified by various metrics also coded within *RamanSPy* (e.g., mean squared error (MSE), spectral angle distance (SAD) (Kruse et al., 1993) and spectral information divergence (SID) (Chang, 1999)), which we use to measure the performance of each denoising method by comparing denoised signals against the provided high SNR data, which act as a reference.

Importantly, applying this pipeline to new data only involves changing the data source. Taking advantage of this transferability, we test the denoiser on unseen volumetric Raman data from another cell line (THP-1 (Kallepitis et al., 2017)), to which we added Gaussian noise. For these data, Figure 2c shows improved denoising performance according to the MSE metric, which is dependent on normalisation and scale, but with lower significance according to scale-invariant metrics available in *RamanSPy*. This example emphasises the sensitivity of algorithms to data shifts and the importance of incorporating robust validation criteria based on the unique requirements of each application. The collection of metrics that *RamanSPy* provides is intended to serve as a starting resource to test performance according to different objectives.

---

[1]Model was trained on spectra from MDA-MB-231 breast cancer cells (Horgan et al., 2021). Deposited by authors at: https://github.com/conor-horgan/DeepeR.

**AI interoperability.** Secondly, the data management backbone of *RamanSPy* ensures a direct data flow to the rest of the Python ecosystem, i.e., data can be loaded, preprocessed, and analysed in *RamanSPy* and then exported to conduct further modelling and analysis elsewhere (Figure 3a). As an example application, we perform AI-based bacteria identification using Raman measurements from 30 bacterial and yeast isolates as provided in Ho et al. (2019) (Figure 3b). After loading and visualising the spectra with *RamanSPy*, we interface the data with the *lazypredict* Python package (Pandala), which allows us to directly benchmark 28 ML classification models (including logistic regression, support vector machines and decision trees) on the task of predicting the bacterial species from a given spectrum. The models were first trained on the provided fine-tuning dataset (100 spectra per isolate) and then tested on the unseen test set of the same size. Our benchmarking analysis in Figure 3c finds logistic regression as the best-performing model, achieving a classification accuracy of 79.63% on the species-level classification task (Figure 3d), and 94.63% for antibiotic treatment classification (Figure 3e).

**Dataset suite for model evaluation.** To further assist the process of testing and evaluating new computational approaches, *RamanSPy* provides access to a library of curated datasets collated from existing literature for different tasks (e.g., classification, denoising, Raman imaging). With this library, we aim to seed the growth of a common repository of RS datasets that helps reduce barriers to data access, especially for ML teams with limited access to RS instruments (Luo et al., 2022). The dataset library in *RamanSPy* already includes data loaders for Raman data from bacterial species (Ho et al., 2019), cell lines (Horgan et al., 2021; Kallepitis et al., 2017), COVID-19 samples (Yin et al., 2021; 2020), multi-instrument Surface Enhanced Raman Spectroscopy (SERS) measurements of adenine samples (Fornasaro et al., 2020), wheat lines (ŞEN et al., 2023), minerals (Lafuente et al., 2015), and will continue to be expanded. Recognizing the potential benefits that synthetic and surrogate datasets can provide for the generation of controlled ground truths for algorithmic validation, we have also integrated the synthetic Raman data generator described in Georgiev et al. (2024) within *RamanSPy*.

## 4. Conclusion

In this paper, we have introduced *RamanSPy* - a computational framework for integrative Raman spectroscopic data analysis aimed at overcoming the limitations of currently available commercial software tools in terms of accessibility, flexibility and reproducibility, and facilitating the adoption and validation of advanced AI technologies for next-generation RS analysis.

The codebase of *RamanSPy* is fully open-source and disseminated under a permissive license that allows for unrestricted use, adaptation, and extension, including for commercial purposes. It is supplemented with extended online documentation containing a comprehensive selection of tutorials and example applications, as well as further information about the analyses presented in this paper. We believe this will be critical for the continuous development of the platform and its adoption across different scientific domains, including biomedical research, chemistry, and materials science, among others. Future directions include the expansion of our suite of built-in methods, tools and datasets; the incorporation of cutting-edge AI technologies into the framework as the field progresses; and the integration of the package into experimental setups and other software solutions.

## Data availability

All data used in this article are previously published open-access data that have been deposited by the respective authors online. Instructions on how to access, download, and load the data sets provided in *RamanSPy* are available in the documentation at https://ramanspy. readthedocs.io/en/latest/datasets.html.

## Code availability

The codebase of *RamanSPy* is open-source and hosted on GitHub at https://github.com/ barahona-research-group/RamanSPy. The package can be installed via pip using 'pip install ramanspy'. Documentation, including detailed tutorials and examples, is available at https://ramanspy.readthedocs.io. The scripts used to produce the analysis results presented in this paper are also provided as executable Jupyter Notebook examples at https://github.com/ barahona-research-group/RamanSPy/tree/ 3dd2c1e09420c5ac473a72ebd6ed06a91c30a85c/ paper_reproducibility and as part of the documentation of *RamanSPy* at https: //ramanspy.readthedocs.io/en/latest/ auto_examples/index.html.

## Acknowledgements

D.G. is supported by UK Research and Innovation [UKRI Centre for Doctoral Training in AI for Healthcare grant number EP/S023283/1]. S.V.P. gratefully acknowledges support from the Independent Research Fund Denmark (0170-00011B). R.X. and M.M.S. acknowledge support from the Engineering and Physical Sciences Research Council (EP/P00114/1 and EP/T020792/1). A.F.-G. acknowledges support from the Schmidt Science Fellows, in partnership with the Rhodes Trust. M.M.S. acknowledges support from the Royal Academy of Engineering Chair in Emerging Technologies award (CiET2021\94). M.B. acknowledges support by the Engineering and Physical Sciences Research Council under grant EP/N014529/1, funding the EPSRC Centre for Mathematics of Precision Healthcare at Imperial College London, and under grant EP/T027258/1. The authors thank Dr Akemi Nogiwa Valdez for proofreading and data management support. Figures were assembled in BioRender.

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
