# OpenReview forum: "RamanSPy: Augmenting Raman Spectroscopy Data Analysis with AI"
_ICML.cc/2024/Workshop/ML4LMS — ML4LMS Poster_

### Official Review · Reviewer_B7om · 2024-05-29
**This paper has already been published in ACS.**

**Rating:** 3
**Confidence:** 5

**Review:**

A quick Google search revealed that this paper has already been published in ACS here https://pubs.acs.org/doi/10.1021/acs.analchem.4c00383 just a couple of weeks ago. I do not think it needs to go through another revision or even needs to be published in another venue.

---

### Official Review · Reviewer_Qspm · 2024-06-09
**Python framework for data analysis of Raman Spectroscopy**

**Rating:** 7
**Confidence:** 3

**Review:**

Raman Spectroscopy is a widely used technique for chemical characterization of solutions of chemical and  biochemical origin and materials. The authors have developed a python package called RamanSPy for data analysis coming from Raman Spectroscopy. The platform emulates several features of the popular machine learning platform, scikit-learn thus allowing a familiar and user friendly experience. The authors create links with several popular machine learning thus enabling loading of pre trained models. It would be good to see if the authors can incorporate AI interpretability and uncertainty quantification techniques into the pipeline. The authors mention about sharing workflows between groups but do not mention about reproducibility.

---

### Official Review · Reviewer_MeLz · 2024-06-12
**This paper serves as an introduction to a source python library, RamanSPy, for streamlining post-Raman Spectroscopy analysis**

**Rating:** 9
**Confidence:** 5

**Review:**

## Summary
The authors observe a lack of data, methodology, software, etc., standards for Raman Spectroscopy. It is also noted that the reproducibility, ease of working, day-to-day activities with the data, and corresponding analysis are still not completely solved in this area. An open-source, modular, extensible library, RamanSPy, is created to address these challenges.
##  Strengths and Weaknesses
The library handles the end-to-end workflow for Raman Spectroscopy from various data formats for the experimental data gathered to processing, analysis, visualisation and metrics.
The library addresses the problem of unifying and standardising the working workflow with Raman Spectroscopy. With a growing trend of using machine learning-based approaches, the library provides support for the use of custom AI models as per users' needs.
The library is well integrated into the broader scientific and machine-learning Python ecosystem through numpy, pandas, scikit-learn, PyTorch, etc.
A survey of any existing libraries for Raman Spectroscopy and other related techniques, experiments and approaches could have been added as part of related/previous works.
## Presentation
The writing is structured very well. The authors have provided detailed figures, comparisons, code samples and experimental results. Further supplementary material to the work exists in the form of the library code, documentation and related tutorials.

## Soundness
4/4 (Excellent)
## Contribution
4/4 (Excellent)